# Lessons Learned from the Testing of Neonatal Vitamin A Supplementation

**DOI:** 10.3390/nu11020449

**Published:** 2019-02-21

**Authors:** Christine Stabell Benn, Peter Aaby, Ane Bærent Fisker

**Affiliations:** 1Research Center for Vitamins and Vaccines, Bandim Health Project, Statens Serum Institut, 2300 Copenhagen, Denmark; p.aaby@bandim.org (P.A.); abf@ssi.dk (A.B.F.); 2OPEN, Institute for Clinical Research, University of Southern Denmark/Odense University Hospital, 5000 Odense, Denmark; 3Bandim Health Project, Bissau, 1004 Bissau Codex, Guinea-Bissau

**Keywords:** neonatal vitamin A supplementation, diphtheria-tetanus-pertussis vaccine, infant mortality, low-income countries, vitamin A deficiency

## Abstract

A total of 12 trials have tested the effect of neonatal vitamin A supplementation (NVAS) on mortality. Overall, NVAS had no effect on mortality, but results were heterogeneous. Two competing hypotheses have been put forward to explain the divergent effects: A) NVAS works by preventing vitamin A deficiency (VAD) and not all countries have VAD; B) NVAS interacts negatively with subsequent diphtheria-tetanus-pertussis (DTP) vaccine, increasing mortality in females; in countries with low DTP coverage NVAS may have a beneficial effect. Only hypothesis A was tested in a recent meta-analysis; there is no strong empirical support for hypothesis A and it would not explain observed negative effects in some settings. Hypothesis B accounts for most observations. However, so far it has only been tested properly in a few trials. If hypothesis B is correct, it has major consequences for the understanding of the effects of vitamin A, and for the VAS policy in older children. As a WHO priority, the DTP coverage is bound to increase, and therefore hypothesis B urgently needs to be tested.

## 1. Introduction

Vitamin A deficiency (VAD) is widespread in low-income settings. In the late 1980s and early 1990s, several randomized trials tested the effect on overall mortality of providing high-dose vitamin A supplementation (VAS) to children between 6 months and 5 years of age. A meta-analysis showed that VAS was associated with 23% reduced child mortality [1], and in 1993 WHO recommended biannual VAS to all children between 6 months and 5 years of age [2]. 

As newborns are born with low vitamin A stores, it seemed obvious to attempt to extend the benefits to younger children. However, studies in children aged 1–5 months of age showed no beneficial effects and even a tendency for negative effects of VAS [3]. 

### 1.1. First NVAS Trials in 1996–2008

Following a randomized trial from Indonesia, showing a beneficial effect of neonatal vitamin A supplementation (NVAS) [4], several other NVAS trials were carried out in Asia and Africa. By 2008, six trials had been conducted [4,5,6,7,8,9,10], showing mixed results. Three trials, in Indonesia [4], India [7], and Bangladesh [8], showed reduction in mortality. Three other trials conducted in Nepal [10], Zimbabwe [5,6] and Guinea-Bissau [9] found no effect or a tendency for negative effects. 

### 1.2. WHO Technical Consultation Regarding NVAS in 2008

In December 2008, WHO convened a technical consultation regarding NVAS [11]. The PIs of the six trials were present. 

Two hypotheses for the conflicting results were discussed:
NVAS was beneficial in Asia, but not in Africa, presumably due to more widespread vitamin A deficiency in the Asian settings (“NVAS-prevents-vitamin-A-deficiency-hypothesis”) [12]. NVAS interacted with vaccines, amplifying both beneficial and deleterious non-specific effects. This would entail that NVAS be beneficial in the first months of life while the live BCG vaccine (provided at birth) was the most recent vaccination, but the effect ceased to be beneficial and became negative for females once they started receiving the inactivated diphtheria-tetanus-pertussis (DTP) vaccine around 1½–2 months of age (“negative-interaction-between-NVAS-and-DTP-in-females-hypothesis”) [3] (Figure 1).

The proponents of hypothesis B recommended conducting a pooled analysis of all existing datasets before undertaking new trials. This pooled analysis should be aimed at understanding the differential effect of NVAS by sex and vaccine status [11]. However, the majority of PI’s declined, stating that they would conduct a pooled analysis of three of the four trials conducted in Asia [11] (by 2018, no results have been presented). 

### 1.3. WHO Workshop Regarding NVAS in 2014

Instead, with Gates Foundation funding, WHO decided to carry out three new mega-trials of NVAS, to “inform global policy on the use of neonatal vitamin A supplementation” [13]. Initially there was no plan to collect vaccination data, but it was subsequently put forward that to test hypothesis B, the trials should be analyzed according to DTP-vaccination status, and this point was included in the published protocol for the three trials [13]. 

When the results of the three new NVAS trials became available in 2014, WHO convened a workshop. It was clear that NVAS was not associated with systematic benefits [14,15,16]. In addition, the results of three other NVAS trials had become available; NVAS had no beneficial effect in any of these trials either [17,18,19].

Here, we review the total evidence to see if there are lessons to be learned from the >20 years of research into NVAS. We conclude that it is urgently necessary for hypothesis B to be tested. 

## 2. Materials and Methods 

Potential NVAS trials for inclusion in the review were identified using the same methods as in a recent meta-analysis [20]. Individual or cluster randomized trials assessing the effect of early NVAS (25,000–50,000 IU intended to be given within the first 2–3 days of life) compared with placebo, with follow-up through at least 6 months of age, were eligible. In addition, we included the Nepal trial [10], which provided NVAS within the first month of life and followed children for 4 months, and which was one of the original six trials assessed at the technical consultation in 2008 [11]. 

## 3. Results

A total of 12 randomized trials assessing the effect of neonatal VAS (NVAS) versus no NVAS/placebo on overall mortality have been carried out; one was reported separately for HIV positive and HIV-negative mothers (Table 1). 

### 3.1. WHO Meta-Analysis

In 2014, WHO arranged an analytical workshop including all the PIs of NVAS trials. Based on 11 trials (excluding the Nepalese NVAS trial [10], which did not provide NVAS in the first days of life) the meta-pooled RR with a random effects model was 0.97 (95% CI 0.89–1.06) by 6 months of age and 1.00 (0.93–1.08) by 12 months of age [20]. There was considerable heterogeneity across trials. Hence, the aim was to conduct a meta-analysis to explore reasons for heterogeneity. Two types of analysis were carried out to explore the reasons for heterogeneity: A *meta-regression* of the 11 trial estimates by study-level characteristics, and a *meta-analysis* of trial-specific estimates stratified by individual-level characteristics [20]. 

Below, we review the evidence pro and con hypotheses A and B, respectively. 

### 3.2. Evidence pro and con Hypothesis A

The WHO meta-regression showed that NVAS was associated with significantly reduced 6-month mortality in the trials conducted in Southern Asia, in contexts with moderate or severe maternal vitamin A deficiency (VAD), where mortality before 6 months was >30/1000 live births, >75% of infant mortality occurred in the first 6 months or >32% mothers had no schooling [20]. Maternal VAD was defined as 10% or more women with serum retinol <0.7 umol/L or 5% or more women with night blindness [20]. Assessing vitamin A status by means of serum retinol has well known limitations [22]. The meta-analysis by individual level characteristics did not support that either maternal VAD was an important effect modifier; for example, there was no beneficial effect of NVAS in children of mothers who had not received vitamin A supplementation, nor was there any beneficial effect of NVAS in children of mothers who suffered from night blindness. Further supporting the theory that the heterogeneous effects of NVAS were not explained by differences in VAD prevalence across studies, there was no beneficial effect in low birth weight or preterm newborns [20], who would be likely to suffer from VAD [23]. In Table 2, we listed the uncontested observations from NVAS trials. Overall, there is limited support for hypothesis A. and notably, hypothesis A cannot explain the many situations in which NVAS was found to have negative effects (Table 2). 

### 3.3. Evidence pro and con Hypothesis B

In the three NVAS trials from Guinea-Bissau, which directly tested the effect of NVAS before and after DTP vaccination, NVAS was associated with increased female mortality after DTP, the NVAS/placebo mortality relative risk in females who had received DTP being 2.19 (1.09–4.38) [24], 1.44 (0.75–2.78) [17] and 1.31 (0.61, 2.81) [25], respectively, the combined estimate being 1.62 (1.08–2.42). 

The India 2014 and the Ghana trials tested the interaction between NVAS and DTP vaccination, but data was not presented by sex. In Ghana, NVAS recipients had 36% (−4–92%) higher mortality after receiving DTP [14]. Conversely, India reported that NVAS recipients had 29% (2–49%) lower mortality after DTP [16] (*p* value for homogeneity = 0.007) (Table 3). 

Unfortunately, the Indian data are flawed. The children were only visited at ages 1 month, 3 months, and 6 months, but virtually all deaths between 1 and 6 months were included in the analysis. Thus, children without information about DTP at 3 and 6 months were classified as DTP-unvaccinated. This would include children who were vaccinated and died between visits. For survivors, vaccine status was updated retrospectively at next visit. This methodology introduces survival bias [28]. Consequently, reported mortality in India among DTP-unvaccinated participants was 324% (255–407%) higher than for the DTP-vaccinated participants, much higher than the 81% observed in Ghana, where vaccination data were collected monthly and survival bias would be less (Table 3). These results are incompatible (*p* value for homogeneity < 0.001) and since there is a survival bias for the Indian data, only the Ghanaian suggestion that NVAS recipients tended to have increased mortality after DTP is valid. 

No other trial tested for NVAS-DTP interaction. All the NVAS trials, which found beneficial effects, had low DTP coverage and/or only followed children to 6 months [29,30]. An ecological analysis of the trials which provided data on mortality from 6–12 months, shows that in females, the effect of NVAS is associated with 20% (2–42%) increase in mortality from 6–12 months, when DTP-vaccine would be the dominating vaccine, further supporting hypothesis B [27] (Table 4).

### 3.4. Biological Considerations

While VAD is associated with increased mortality, and it makes intuitive sense that providing VAS would reduce mortality by preventing and treating VAD, there is in fact limited evidence to support that VAS works by preventing VAD; there is no clear association between the degree of preexisting VAD and the effect of VAS to older children and distribution of high-dose VAS turns out to have limited effect on vitamin A status [29,31,32]. 

Vitamin A has profound importance for the immune system [33]. Human in vitro studies have shown that vitamin A Th1 and Th2-related chemokine expression [34] and induced innate tolerance in human monocytes [35]. In a very recent human experiment, DTP vaccine was associated with induction of innate tolerance in females (Blok et al., in revision). The combination of vitamin A and DTP vaccine was associated with significantly increased parasitaemia levels in female mice in an experimental murine malaria model [36]. Hence, there is some biological evidence to support a negative effect of combining VAS and DTP vaccine in females. 

## 4. Discussion

### 4.1. Current Status with Regard to NVAS

The hypothesis, which best explains the totality of data, including the heterogeneity between NVAS trials, is hypothesis B: negative interaction between NVAS and DTP in females. Nonetheless, the majority of PIs in the 2014 workshop wished to emphasize that NVAS had beneficial effects in the Asian countries where there was maternal VAD but no beneficial effect in the African countries where there was no maternal VAD as support for hypothesis A [20]. The invitation to analyze the data for a potential interaction between NVAS and DTP in females was declined, as it had been in 2008. As a consequence, the proponents of hypothesis B withdrew from the meta-analysis paper [20].

As of today, hypothesis B about a negative interaction between NVAS and DTP-containing vaccines in females has not been tested in the new NVAS trials, or in any of the other NVAS trials, except the Guinean trials. The PIs of the other NVAS trials have declined to share their data for that purpose. 

WHO is still considering whether NVAS should be recommended at least in certain parts of the world. The overall effect of NVAS may be beneficial in sub-populations as long as DTP-coverage is low, and where the post-neonatal mortality is very low compared with the neonatal mortality. However, if hypothesis B is correct, NVAS could cause harm if implemented in areas with increasing or high DTP coverage and with a lower neonatal/post-neonatal mortality ratio. Importantly, there would be no way of knowing since it would be considered unethical to conduct randomized trials after the implementation. 

### 4.2. Lessons Learned

From the first NVAS trial was published in 1996 until 2008, six NVAS trials were conducted. Results were heterogeneous. At a WHO technical consultation, it was decided to carry out three mega-trials to inform global policy. Now, 10 years later, we have 12 trials – and results are still heterogeneous. None of the new trials showed any benefit of NVAS by 12 months. 

Two competing hypotheses for the heterogeneous results have been put forward. One (A) is intuitive and compelling since it fits with current assumptions about VAS as prevention and treatment of VAD – but is not supported by data (Table 2). The other (B) challenges current assumptions but is supported by the available data (Table 4) – but it has not been possible to get others to test it. 

In our opinion, this situation: that we have spent millions of dollars and we are still not sure about the potential benefits or harms of NVAS and what is causing the heterogeneity, could have been avoided as documented below. 

We believe there are several important lessons to be learned from the NVAS story:I.***When results are heterogeneous, use existing data to explore reasons for heterogeneity prior to conducting new studies***

By 2008, both the India [7], the Bangladesh [8] and the Zimbabwe [5,6] trials had data which could be used to test hypothesis B. It may be that the results of this testing could have provided answers, which had made the three new trials either superfluous or more informative.
II.***In a situation with two competing hypotheses, it is important to compare their ability to explain the totality of observations. Preference should be given to pursue the hypothesis which best explain the data***

In the choice between an intuitive hypothesis (A) that NVAS works in settings with VAD, and the heterogeneity in effects was due to differences in maternal VAD, and a competing and dogma-challenging hypothesis (B) that NVAS interacts negatively with DTP vaccine in females, preference was given to hypothesis A in the WHO meta-analysis [20]. Hypothesis B has not been tested; reasons for not doing so have not been provided. However, as argued here, hypothesis B provides the most coherent explanation for the heterogeneity between trials, and it should be further tested.
III.***When in the possession of data to test a plausible hypothesis, data should be made available for testing the hypothesis***

There should be no excuse for not sharing data to test a plausible hypothesis with potential consequences for millions of children. In the case of NVAS, the lack of data sharing may have been very expensive. Had evidence been found in support of hypothesis B in 2008, the three new trials might not have been needed, and millions of dollars could have been saved. It would also have led to consideration of the potential negative interactions between VAS and DTP in older children at an earlier stage. Once the new trials, which specifically collected data on vaccination status to explore the hypothesis of interaction between NVAS and DTP, had been conducted, it is hard to understand why the PIs did not test hypothesis B. Most of the NVAS trials received funding from Bill and Melinda Gates Foundation, which emphasizes data sharing. 

## 5. Conclusions

The hypothesis that NVAS interacts negatively with subsequent DTP vaccine in females is relevant not only for neonates, but also because VAS to older children is often given together with or close to a DTP vaccine: Many children receive their primary DTP series with delays and many countries recommend DTP-boosters. Thus, if VAS interacts negatively with DTP in females, it could have negative consequences for older children. Many studies have indeed suggested that it is harmful to combine VAS and DTP in older children [25,30,37,38,39,40,41]. This should be of immediate concern in relation to the VAS policy to older children. 

In hindsight, we might have been able to save millions of dollars and numerous lives, had the NVAS community been able to collaborate. It is still possible to correct some of these mistakes if hypothesis B is tested in all available datasets now. The NVAS trials from India 2003, Ghana, Tanzania have good data to do this; Bangladesh may have data for a landmark analysis, and age group analyses can be done using the Zimbabwe dataset. An analysis plan has been sent to the PIs of all NVAS trials and to WHO on numerous occasions and is provided at [42]. We reiterate the call for urgent testing of the hypothesis and hope that the lessons learned in this process may inform future evaluations of interventions. 

## Figures and Tables

**Figure 1 nutrients-11-00449-f001:**
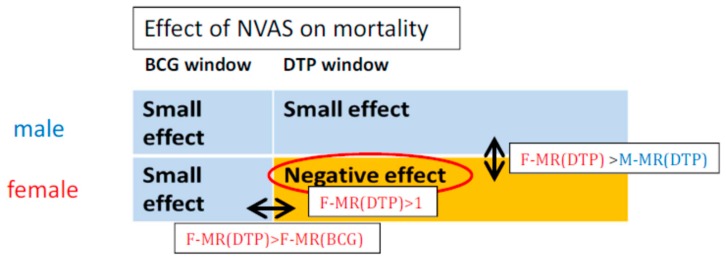
Illustration of Hypothesis B and its derived deductions. Note: F-MR(BCG) signifies NVAS/placebo mortality ratio in BCG-vaccinated females; F-MR(DTP) and M-MR(DTP) signifies NVAS/placebo mortality ratio in DTP-vaccinated females and males, respectively.

**Table 1 nutrients-11-00449-t001:** Main results of randomized trials of the effect of neonatal vitamin A supplementation on overall mortality by 6 and 12 months of age.

Trial (Authors, Country, Year of Publication)	NVAS vs. Placebo Relative Risk of Mortality (95% CI) 6 Months of Age	NVAS vs. Placebo Relative Risk of Mortality (95% CI) 12 Months of Age
West et al, Nepal, 1995 [10]	1.07 (0.66–1.72)	N/A
Humphrey et al, Indonesia, 1996 [4]	0.39 (0.16–0.93) ^1^	0.36 (0.16–0.87)
Rahmathullah et al., India, 2003 [7]	0.78 (0.63–0.96)	N/A
Malaba et al, Zimbabwe, 2005 [5]/Humphrey et al, Zimbabwe, 2006 [6]	0.98 (0.84–1.13) ^1^	1.05 (0.93–1.19) ^1,2^
Benn et al, Guinea-Bissau, 2008 [9]	1.16 (0.78–1.73) ^1^	1.07 (0.79–1.44)
Klemm et al, Bangladesh, 2008 [8]	0.85 (0.73–1.00)	0.85 (0.73-0.99) ^1^
Benn et al, Guinea-Bissau, 2010 [17]	1.02 (0.73–1.44) ^1^	1.08 (0.79–1.47)
Benn et al, Guinea-Bissau, 2014 [18]	1.10 (0.75–1.61)^1^	1.28 (0.91–1.81)
Edmond et al, Ghana, 2014 [14]	1.12 (0.95–1.33)	1.13 (0.98–1.31)
Masanja et al, Tanzania, 2014 [15]	1.10 (0.95–1.26)	1.04 (0.93–1.17)
Mazumbder et al, India, 2014 [16]	0.90 (0.81–1.00)	0.94 (0.86–1.02)
Soofi et al, Pakistan, 2016 [19]	1.06 (0.82–1.37)	1.00 (0.93–1.08) ^1^

^1^ Obtained from [20]. ^2^ The trial was a 2×2 factorial trial with maternal supplementation. Maternal supplementation is no longer recommended by WHO [21]. The effect estimates in children of unsupplemented mothers was 1.18 (0.76–1.83) [5] and 1.21 (0.99–1.46) [6] in children of HIV negative and HIV positive mothers, respectively.

**Table 2 nutrients-11-00449-t002:** The ability to explain current evidence with Hypothesis A: Neonatal-vitamin A-supplementation-prevents-vitamin-A-deficiency.

Observation from NVAS Trials	Hypothesis A: NVAS-Prevents-Vitamin-A-Deficiency-Hypothesis
NVAS positive effect in some Asian studies up to 6 months of age [20]	FITS: Maternal VAD more prevalent in the Asian studies
NVAS negative effect in African studies, the meta-estimate being 1.07 (1.00–1.15) [20]	DOES NOT FIT: Several of the African studies were conducted in low-birth weight infants [17], and in children with low vitamin A stores [6,26]; yet no beneficial effect was found. Lack of maternal VAD would not explain why the effect would be negative in Africa.
Sex-differential NVAS effects from 6–12 months; negative effect in females from 6–12 months [27]	DOES NOT FIT: The hypothesis cannot explain negative effects in females from 6–12 months
NVAS harmful after DTP [14]	DOES NOT FIT: The hypothesis cannot explain negative effects after DTP
NVAS harmful in females after DTP in the three studies which analyzed it [17,18,24]	DOES NOT FIT: The hypothesis cannot explain negative effects in females
NVAS beneficial in areas with maternal VAD [20]	FITS
NVAS beneficial in areas with high early infant mortality [20]	FITS if one assumes that high early infant mortality is a proxy for VAD.
NVAS beneficial in areas with low maternal schooling [20]	FITS if one assumes that low maternal schooling is a proxy for VAD. DOES NOT FIT with the fact that NVAS was not more beneficial in children of mothers who had not received VAS/mothers who suffered from night blindness (next row)
No beneficial NVAS effect in children of mothers, who had not received vitamin A or who suffered from night blindness [20]	DOES NOT FIT: children of mothers, who had not received vitamin A or who suffered from night blindness would be the most likely to suffer from VAD
No beneficial NVAS effect in low birth weight or preterm newborns [20]	DOES NOT FIT: low birth weight or preterm newborns would be the most likely to be suffering from VAD

**Table 3 nutrients-11-00449-t003:** The effect of neonatal vitamin A supplementation in relation to first dose of DTP vaccine in the neonatal vitamin A supplementation (NVAS) trials.

	DTP-Unvaccinated vs. Vaccinated Relative Risk of Mortality (95% CI)	NVAS vs. Placebo Relative Risk of Mortality (95% CI)	*p* for Homogeneity
Ghana [15]	1.81 (1.34–2.43)		
Before receiving DTP-HepB-HiB ^1^		0.81 (0.50–1.32)	
After receiving DTP-HepB-HiB ^1^		1.36 (0.96–1.92)	0.089
India [16]	4.24 (3.55–5.07)		
Before receiving DTP		0.92 (0.78–1.07)	
After receiving DTP		0.71 (0.51–0.98)	0.167

^1^ DTP-HepB-HiB = diphtheria-tetanus-pertussis-Hepatitis B-*H. Influenzae* type b vaccine.

**Table 4 nutrients-11-00449-t004:** The ability to explain current evidence with Hypothesis B: Negative-interaction-between-NVAS-and-DTP-in-females.

Observation from NVAS Trials	Hypothesis B: Negative-Interaction-between-NVAS-and DTP-in-Females
NVAS positive effect in some Asian studies up to 6 months of age [20]	FITS: DTP vaccine was given late and with lower coverage in the Asian studies [16,30] and negative effects were not yet apparent by 6 months of age
NVAS negative effect in African studies, the meta-estimate being 1.07 (1.00–1.15) [20]	FITS: DTP was given early and with high coverage in the African studies [14,15,30]
Sex-differential NVAS effects from 6–12 months; negative effect in females from 6–12 months [27]	FITS: A negative effect interaction between NVAS and DTP in females would become visible after children started receiving DTP, i.e., it would be more prominent from 6–12 months
NVAS harmful after DTP [14]	FITS
NVAS harmful in females after DTP in the three studies which analyzed it [17,18,24]	FITS
NVAS beneficial in areas with maternal VAD [20]	FITS since DTP was given later and with lower coverage in the trials with maternal VAD
NVAS beneficial in areas with high early infant mortality [20]	FITS as in areas with high early infant mortality, most deaths would occur before DTP
NVAS beneficial in areas with low maternal schooling [20]	FITS if one assumes that low maternal schooling was a proxy for later DTP and lower coverage of DTP
No beneficial NVAS effect in children of mothers, who had not received vitamin A or who suffered from night blindness [20]	FITS if major determinant of NVAS effect is timing of DTP vaccine rather than VAD
No beneficial NVAS effect in low birth weight or preterm newborns [20]	FITS if major determinant of NVAS effect is timing of DTP vaccine rather than VAD

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
