# Peer review of "Lessons Learned from the Testing of Neonatal Vitamin A Supplementation"

_nutrients, 2019, doi:10.3390/nu11020449_

Round 1
Reviewer 1 Report
Benn et al., Nutrients 2019
General:
Although this paper is labelled as a “review” it is more or less an opinion paper. This reviewer is unsure if Nutrients has an appropriate category for this type of article. For the most part, this reviewer is in agreement with the lack of basic work that has gone into the neonatal vitamin A supplementation studies before the experimenting on 1000’s of children. The authors raise a concern about the safety of supplementation to older children when vaccinations are given in close proximity. This is a huge question mark considering millions of children who are receiving vitamin A supplements each year.
The paper needs more details for those readers who are outside of the field and have not been following this argument.
The method of measuring vitamin A status in the various trials needs to be mentioned. Serum retinol is not a good measure and this caveat needs to be in the paper. We really know very little about the vitamin A status in any of the large trials.
Specific:
Line 15: Insert “a” before beneficial.
L. 20: Delete “also”
L. 34: Delete “very”
L. 54: Delete “the”
L. 59: Missing an “s” on “mega-trials”
L. 63: Delete “new”
Table 1: More detail is needed in this table to be understandable. What are the main results? Are these relative ratios of mortality? There is a spacing issue mid-way down the table.
In the footnote, it is state “Maternal supplementation is not recommended.” By whom and is there a reference?
Table 2: How was the maternal vitamin A status measured especially in relationship to VAD?
What does it mean “DOES NOT FIT at the individual level? VAD and its relationship to years of school? Is this because of the limitations of using serum retinol concentration as the biomarker? The limitations of vitamin A biomarkers needs to be discussed in relationship to Hypothesis A somewhere in the article, whether it be introduction or discussion.
No “,” needed between “mothers who had not…”
In the final row of the Table, somewhere in the manuscript the biology of the liver in low birthweight and preterm infants needs to be discussed in the paper. The liver is not fully functioning in preterm infants. That could be a major problem in this group of infants.
Author Response
General:
Although this paper is labelled as a “review” it is more or less an opinion paper. This reviewer is unsure if Nutrients has an appropriate category for this type of article. For the most part, this reviewer is in agreement with the lack of basic work that has gone into the neonatal vitamin A supplementation studies before the experimenting on 1000’s of children. The authors raise a concern about the safety of supplementation to older children when vaccinations are given in close proximity. This is a huge question mark considering millions of children who are receiving vitamin A supplements each year.
CB: We are happy that the reviewer agrees with us on these points
The paper needs more details for those readers who are outside of the field and have not been following this argument.
CB: We have added more detail, and inserted subtitles, which hopefully assists the readers.
The method of measuring vitamin A status in the various trials needs to be mentioned. Serum retinol is not a good measure and this caveat needs to be in the paper. We really know very little about the vitamin A status in any of the large trials.
CB: This information has been added
Specific:
Line 15: Insert “a” before beneficial.
CB: Done
L. 20: Delete “also”
CB: Done
L. 34: Delete “very”
CB: Done
L. 54: Delete “the”
CB: Done
L. 59: Missing an “s” on “mega-trials”
CB: Done
L. 63: Delete “new”
CB: Done
Table 1: More detail is needed in this table to be understandable. What are the main results? Are these relative ratios of mortality? There is a spacing issue mid-way down the table.
CB: More detail has been added. The spacing issue is due to one of the trials being reported in two separate papers. We have inserted table lines to make this clearer.
In the footnote, it is state “Maternal supplementation is not recommended.” By whom and is there a reference?
CB: By WHO – this and a reference has been added.
Table 2: How was the maternal vitamin A status measured especially in relationship to VAD?
CB: The information has been added in the text, and a reference is provided in tables 2 and 3.
What does it mean “DOES NOT FIT at the individual level? VAD and its relationship to years of school? Is this because of the limitations of using serum retinol concentration as the biomarker? The limitations of vitamin A biomarkers needs to be discussed in relationship to Hypothesis A somewhere in the article, whether it be introduction or discussion.
CB: We have tried to clarify in the text under hypothesis A and in table 2 what we find “DOES NOT FIT”. We have added a sentence about the limitation of the vitamin A biomarkers.
No “,” needed between “mothers who had not…”
CB: Changed as suggested.
In the final row of the Table, somewhere in the manuscript the biology of the liver in low birthweight and preterm infants needs to be discussed in the paper. The liver is not fully functioning in preterm infants. That could be a major problem in this group of infants.
CB: Without references we are a bit unsure about what the reviewer means. We have inserted a reference to the statement that the liver stores of low-weight and preterm newborns are expected to be lower, and if VAS worked by preventing VAD, one would expect an effect in these children.
Reviewer 2 Report
This is an important paper that raises some significant lessons for the community of researchers in perinatal nutrition.
The work is well described and the analysis is performed with rigor.
In the discussion and conclusions the authors language deteriorates into wording that makes this paper read like a personal grudge instead of a call to action. I request that the authors review this and revise accordingly.
The paper would also be strengthened by adding physiologic processes that may explain why NVAS after DPT vaccination may have a harmful interaction. Is this an inflammatory mediated process?
Addressing these points in addition to some minor editing of English language (worse towards the end of the paper) will significantly improve the paper.
Author Response
This is an important paper that raises some significant lessons for the community of researchers in perinatal nutrition.
The work is well described and the analysis is performed with rigor.
CB: We thank the reviewer for this positive feed-back
In the discussion and conclusions the authors language deteriorates into wording that makes this paper read like a personal grudge instead of a call to action. I request that the authors review this and revise accordingly.
CB: We have tried to tone down the language.
The paper would also be strengthened by adding physiologic processes that may explain why NVAS after DPT vaccination may have a harmful interaction. Is this an inflammatory mediated process?
CB: We are happy for these request and have added more about the potential biological mechanisms.
Addressing these points in addition to some minor editing of English language (worse towards the end of the paper) will significantly improve the paper.
CB: We have carefully reviewed the English language; as not native in English some errors may remain, we apologise in advance.
Round 2
Reviewer 1 Report
Benn et al., Nutrients 2019
General:
The paper has removed much of the prose that made it opinion. The paper sheds light on the importance of data sharing and maximizing outcomes of expensive trials.
Line 108: mmol should be micromole with the Greek letter mu
Line 114: “of” before NVAS
L. 156: Delete “ly” on intuitive
L. 172: The word “Nonetheless” is hanging. Is it meant for the next sentence?
Author Response
Thank you for the positive feed-back. All the requested edits have been made.
Reviewer 2 Report
Section 3.4 line 162 needs clarification
Section 4.1 line 172 However should be deleted
Author Response
Revised as suggested (I interpret the "however" as the "nonetheless", which was hanging, as pointed out by reviewer 1).